# Soil Microbial Co-Occurrence Patterns under Controlled-Release Urea and Fulvic Acid Applications

**DOI:** 10.3390/microorganisms10091823

**Published:** 2022-09-12

**Authors:** Zeli Li, Kexin Zhang, Lixue Qiu, Shaowu Ding, Huaili Wang, Zhiguang Liu, Min Zhang, Zhanbo Wei

**Affiliations:** 1National Engineering Laboratory for Efficient Use of Soil and Fertilizer Resources, College of Resources and the Environment, Shandong Agricultural University, Taian 271018, China; 2Shandong Wanhao Fertilizer Co., Ltd., Jinan 251600, China; 3Institute of Applied Ecology, Chinese Academy of Sciences, Shenyang 110000, China

**Keywords:** nitrogen, synergist, soil bacterial community, nutrition change, ecological network

## Abstract

The increasing amount of agricultural applications of controlled-release urea (CRU) and fulvic acids (FA) demands a better understanding of FA’s effects on microbially mediated nitrogen (N) nutrient cycling. Herein, a 0–60 day laboratory experiment and a consecutive pot experiment (2016–2018) were carried out to reveal the effects of using CRU on soil microbial N-cycling processes and soil fertility, with and without the application of FA. Compared to the CRU treatment, the CRU+FA treatment boosted wheat yield by 22.1%. To reveal the mechanism of CRU+FA affecting the soil fertility, soil nutrient supply and microbial community were assessed and contrasted in this research. From 0–60 days, compared with the CRU treatment, leaching NO_3_^−^-N content of CRU+FA was dramatically decreased by 12.7–84.2% in the 20 cm depth of soil column. Different fertilizers and the day of fertilization both have an impact on the soil microbiota. The most dominant bacterial phyla *Actinobacteria* and *Proteobacteria* were increased with CRU+FA treatment during 0–60 days. Network analysis revealed that microbial co-occurrence grew more intensive during the CRU+FA treatment, and the environmental change enhanced the microbial community. The CRU+FA treatment, in particular, significantly decreased the relative abundance of *Sphingomonas*, *Lysobacter* and *Nitrospira* associated with nitrification reactions, *Nocardioides* and *Gaiella* related to denitrification reactions. Meanwhile, the CRU+FA treatment grew the relative abundance of *Ensifer*, *Blastococcus*, and *Pseudolabrys* that function in N fixation, and then could reduce NH_4_^+^-N and NO_3_^−^-N leaching and improve the soil nutrient supply. In conclusion, the synergistic effects of slow nutrition release of CRU and growth promoting of FA could improve the soil microbial community of N cycle, reduce the loss of nutrients, and increase the wheat yield.

## 1. Introduction

Nitrogen (N) is an essential nutrient for plants, and a massive amount of N fertilizers are applied to the soil to maintain enough food in the world. However, supplying N at rates exceeding the requirements of the crop often results in tremendous N waste and low N use efficiencies [1]. This has resulted in adverse effects such as plant diseases, soil acidification, agricultural non-point source pollution, and greenhouse gas emissions [2]. An important need for modern agriculture is to develop alternative agrochemical technology to sustain high yield while protecting the environment. To this end, controlled-release urea (CRU) has been developed as an alternative for the conventional urea while fulvic acids (FA) have been applied as a fertilizer synergist to enhance N use efficiency [3].

CRU releases N at a pace that nearly matches crop nutrient absorption when compared with the conventional N chemical fertilizer such as urea [4]. In addition to increasing crop yield, a single application of CRU can save time and labor compared with multiple applications of conventional N fertilizer [5]. Zhao [6] discovered that starch–castor oil-containing superabsorbent polyurethane-coated urea improved water absorption and release performance. This research is crucial for the technological advancement of controlled-release fertilizers. However, CRU has been deemed too expensive for application in crops, particularly in underdeveloped nations. Science-based recommendations on how CRU can be best applied to optimize its performance is still lacking for wide-scale agronomic applications.

Fulvic acids contain about 50% carbon, 45% oxygen, and less than 1% N, with very low levels of phosphorus (P), potassium (K), and other nutrients [7]. As a result, using FA alone did not directly feed plants with important critical nutrients [8]. However, fulvic acid is made up of a variety of aliphatic and aromatic structures with several distinct functional (mostly oxygen-containing) groups that retain nutrient elements in the soil through cation exchange, chelation, complexation, and adsorption [9]. Other functionalities of fulvic acids include improving soil physical properties by promoting the formation of soil aggregates [10], enhancing plant physiological characteristics such as increasing photosynthesis and reducing transpiration [11,12], increasing plant tolerance of environmental stresses [13], stimulating membrane stability and enzyme activity related with N metabolism [14], and ultimately increasing crop growth and yield [15,16]. 

Realizing the benefits and limitations of CRU and FA, we have recently extended our research to explore their synergy when being applied together [17]. Our results indicated that CRU combined with FA significantly increased the levels of nitrate reductase, glutamate dehydrogenase, and auxin in the root and leaf of maize along with enhanced C/N metabolic process compared with CRU alone. Microorganisms are the engine that drives the biogeochemical cycle of soil elements. N addition directly affects the community composition, activity, and metabolic rate of microorganisms by changing the availability of soil N, which then affects soil organic matter transformation and the carbon and N cycle of the ecosystem. The soil microbial communities are an important factor affecting N cycling and N use efficiency; however, the impact of CRU combined with FA on soil microbial communities has not been reported.

Given that the slow release of nutrients by CRU can enhance the soil fertility environment [18,19] and FA modifies soil properties, boost crop root growth, and increases the multiplication of beneficial soil microorganisms [20,21], we integrated the results of a microbial incubation test, a soil column leach experiment, and a greenhouse pot study. Our specific objectives are to (1) investigate the temporal changes in the soil microbial community under the influence of CRU combined with FA, (2) elucidate the linkage between microbial taxa associated with the soil N and soil environment rendered by CRU combined with FA, and (3) evaluate soil microbial factors as they relate to crop yield and N use efficiency with CRU combined with FA.

## 2. Materials and Methods

### 2.1. Study Site and Materials

The soil sample was obtained from the top 20 cm depth from a research farm (36°9′40″ N, 117°9′48″ E) in Shandong Province, China, which was classified into Typic Hapli-Udic Argosol in Chinese soil classification. The soil specimen was air dried and passed through a 10-mesh sieve for further experimentation. The soil had a pH of 7.85 (with soil to water ratio of 1:2.5), 12.01 g kg^−1^ organic matter content, 0.65 g kg^−1^ total N, 72.35 mg kg^−1^ nitrate-N, 9.44 mg kg^−1^ ammonium-N, 13.22 mg kg^−1^ Olsen-P, and 92.22 mg kg^−1^ exchangeable K. With an average annual temperature of 13 °C, the location has a typical continental monsoonal climate. The mean soil temperature was about 9.2 °C during the 2017–2018 wheat growing season. The total irrigation and annual precipitations were 200 mm and 152.6 mm, respectively (Appendix A). The study used a resin-coated CRU (with three-month release period, 43% N) and an FA powder with 5.4 pH 2% N and 3% K_2_O. Other fertilizers used in this study included urea, diammonium phosphate, triple superphosphate, and potassium chloride. Fertilizers and FA powder were provided by Shandong Quanlin Jiayou Fertilizer Company Ltd., Shandong, China.

### 2.2. The Soil Microbial Incubation Experiment

The experiment examined soil microbial community changes among three treatments with three replicates: (1) no N (control); (2) controlled-release urea (CRU); (3) CRU with FA (CRU+FA). Soil samples were mixed with CRU and FA at rates equivalent to field applications (CRU, 450 kg N ha^−1^ and FA, 90 kg ha^−1^). Plastic pots of 8 cm depth were autoclaved and the soil sample was placed at 6 cm depth. Soil moisture was maintained at 30%, which is about 70% of the soil water holding capacity. The soils were incubated at 25 °C in an indoor constant temperature incubator. Soil samples were obtained via destructive sampling on days 1, 7, 30, and 60, and each treatment was sampled to obtain three replicates at each sampling time. Part of the fresh soil sample was used for the microbial 16S rRNA sequencing. The remaining soil sample was used for the measurement of soil ammonia, nitrate N content, and soil pH (Appendix A).

### 2.3. The Soil Column Leaching Experiment

Four treatments, the control, FA, CRU, and CRU+FA experiment was subject to column leaching test (equivalent to CRU, 450 kg N ha^−1^ and FA, 90 kg ha^−1^). A section of PVC pipe (55 cm high and 7.0 cm inner diameter) was sealed with 80-mesh filter cloth along with 20 g sand in the bottom (Appendix A). For each treatment, 1000 g soil was packed to the column for 20 cm depth to reach a field bulk density of 1.29 kg m^−3^. The soil was then covered with about 20 g sand to avoid disrupting the soil layer when water was added. Water was slowly added to reach a moisture content of 30%. On days 1, 7, 30 and 60, 772 mL of water was uniformly applied to the soil column at a rate of 200 mL min^−1^. This amount of water is equivalent to a major irrigation event in the study area [22]. Leachate was collected with a graduate cylinder and analyzed for NH_4_^+^-N, NO_3_^−^-N, bicarbonate (HO_3_^−^), and pH (Appendix A). Soil moisture content was estimated by the amount of water kept in the soil column. Three replicates were conducted for this experiment.

### 2.4. The Pot Experiment

The pot experiment included three treatments with four replicates: (1) no N (control), (2) controlled-release urea (CRU), and (3) controlled-release urea with FA (CRU+FA). A clay pot (36 cm depth and 30 cm diam) was packed with topsoil (around 20 kg) with sand (1 kg) added at the base for drainage (Appendix A). Fertilizers were applied at rates of N 450 kg ha^−1^, P_2_O_5_ 300 kg ha^−1^, and K_2_O 150 kg ha^−1^. The FA was implemented at a rate of 90 kg ha^−1^. Wheat (*Triticum aestivum* L., ‘Jimai 22′) was planted on 12 October 2017, with 45 grains of wheat seeded in each pot (germination rate is 98%, sown at 150 kg hm^−2^), and wheat plants were afterwards intercropped to 36 grains in each pot. With an initiative drip irrigation system, soil moisture was kept at 75% of its water-holding capacity [23]. Soil examples were sampled from 0–20 cm depth on days 33 (seedling stage), 148 (re-greening stage), 174 (jointing stage), and 204 (booting stage) after fertilization for analyses of soil pH, nitrate N, and ammonia N contents (Appendix A). Local producers performed weed and pest control by established growth techniques. At harvesting on 4 June 2018, 200 grains were casually picked from harvested grain, and then weighed to calculate 1000-grain. Plant samples were processed in the laboratory by first deactivating enzymes in an oven for 30 min at 105 °C, then drying at 75 °C to a constant weight, weighing, and grinding the materials to pass through a 100-mesh filter.

### 2.5. Chemical Analyses

A pH meter (PB-10, Sartorius AG, Göettingen, Germany) was used to determine the pH of the soil and fulvic acid using water extraction without CO_2_ (soil: water was 1:2.5). Soil NO_3_^−^-N and NH_4_^+^-N of inorganic N was extracted by 0.01 M CaCl_2_ (soil: water was 1:10) and determined using a succession flow injection analyzer (AA3-A001-02E, Norderstedt, Germany). The Kjeldahl method was used to determine total N [24]. Release characteristics of CRU were measured both in the laboratory with water at 25 °C, and by burying soil bags in the field [19]. The infrared spectrum of FA (Appendix A) was measured by an infrared spectrometer (Nicolet Nexus 410, Waltham, MA, USA).

### 2.6. Molecular Sequence Analysis

To extract DNA from the soil microbial incubation experiment, a soil DNA Extraction Kit (Omega Bio-Tek D5625, Norcross, GA, USA) was employed. An ultraviolet (UV) spectrophotometer was used to evaluate the concentration and purity of the isolated DNA (Thermo NanoDrop2000, Waltham, MA, USA). The widespread primer pairs 515F (5′-GTGCCAGCMGCCGCGG-3′) and 907R (5′-CCGTCAATTCMTTTRAGT-3′) were used to amplify the V4–V5 area of the bacterial 16S phosphorylated RNA gene. In a 20 μL compound inclusion 5 × FastPfu Buffer (4 μL), 2.5 mM dNTPs (2 μL), 5 uM per primer (0.8 μL), 2.5 U μL^−1^ FastPfu Polymerase (0.4 μL), and template DNA (10 ng), PCR reactions were performed in triplicate. The following PCR settings were used: 95 °C (3 min), followed by 27 cycles at 95 °C (30 s), 55 °C (30 s), 72 °C (45 s), and a final extension at 72 °C (10 min) in a thermocycler (ABI GeneAmp 9700, Foster City, CA, USA). Amplicons were quantified using QuantiFluor TMST (Promega, Madison, WI, USA) and sequenced using conventional methods on an Illumina MiSeq platform.

### 2.7. Soil, Water, and Yield Data Analyses

The amount of N in wheat was calculated by building the dry weight of each above-ground wheat portion by the N concentration. The nitrogen uses efficiency (NUE), physiological efficiency (NPE), and agronomic efficiency (NAE) of N were determined using the method described by Devkota [25]:NUE (%) = [(AN − AN0)/(SN)] × 100%
NAE (kg kg^−1^ of N) = (GN − GN0)/SN
NPE (kg kg^−1^ of N) = (YN − YN0)/(AN − AN0).

AN, AN0, and SN represent accumulated N absorb by wheat with N treatment, accumulated N absorb by wheat without N, and total N from fertilizer in the N treatment, respectively; GN and GN0 represent grain dry weight with N treatment and grain dry weight without N treatment, respectively; YN and YN0 represent yield with N treatment and yield without N treatment, respectively.

Data were evaluated by ANOVAs, and mean separations were performed using Duncan tests (*p* < 0.05), all in SAS Version 9.2 (2012, SAS Institute, Cary, NC, USA). The data were processed in Microsoft Excel 2010, and the figures were created in SigmaPlot 2010 (Version 12.0, Systat Software Inc., San Jose, CA, USA).

### 2.8. Soil Microbial Co-Occurrence Network Analysis

The number of 16S rRNA gene sequences from each sample were rarefied to 20,000, which still yielded an average Good’s coverage of 99.09%, respectively. Operational Units (OTUs) were grouped at 97% analogy to conduct statistical analysis of biological data from bacteria using UPARSE (version 7.1). The similarity among the microbial communities in different samples was determined by principal coordinate analysis (PCoA) based on Bray–Curtis dissimilarity using Vegan v2.5-3 package.

To investigate the relationships between taxa within treatments, 36 samples of sequencing data were used to generate taxon co-occurrence patterns and network analyses as follows. The Spearman correlation was calculated for selection of genera with an average relative abundance > 1% within different treatments. The control process of Javanmard’s false discovery rate (FDR) was used to alter all p-values with cutoffs set at 0.001 [26]. Using the random matrix theory, the cutoff for correlation coefficients was established at 0.72. The network was constructed using the updated random matrix in the R environment. The network was visualized using Gephi (http://gephi.github.io/; accessed on 6 June 2021). 

The relationship between the relative abundance of modules in the co-occurrence networks and environmental factors (soil and water properties) was examined by correlation analyses. The module that was most positively correlated with environmental factors was identified and OTU association within the module was displayed with a Manhattan plot calculated using the R environment (http://www.r-project.org; accessed on 6 June 2021). Environmental variables are compared pairwise with a color gradient representing Spearman’s correlation coefficient. Taxonomic classification is based on the following method: Mantel tests were used to determine the relationship between diversity and OTUs of different possible taxa and each environmental condition. The width of the line corresponds to Mantel’s r statistic for the relevant distance correlations, and the color of the edge represents the statistical significance based on 9999 permutations. The correlation map was drawn with Tutool’s platform (https://www.cloudtutu.com; accessed on 6 June 2021).

## 3. Results

### 3.1. Effects of CRU with FA on Soil Bacterial Community Structure

At the OTU level, principal coordinates analysis (PCoA) based on the weighted UniFrac distance matrix identified two main coordinates (PC1 = 9.42% and PC2 = 5.54%), showing clear stratification of soil microbial structure of the control, CRU, and CRU+FA treatment on the 2D domain (Figure 1A). Applications of CRU and CRU+FA clearly affected the soil microbial community structure. These effects can be further seen in PCoA results with the sampling date or duration of incubation identified (Figure 1B). The microbial communication of the control did not show much changes over the course of incubation with negative loadings on both PC1 and PC2. In contrast, clear serial effects of the duration of incubation were observed for CRU and CRU+FA treatments (ANOSIM, r = 0.80, *p* = 0.001), whereby the change in community structure moved up nearly vertically along PC2 from day 1 to day 7, and then sloped down to the lower right quadrant in the 2D domain with positive loading on PC1 and negative loading on PC2 for day 60 (Figure 1B).

The relative abundance of bacteria varied among the CRU and CRU+FA treatments and with days after fertilization at the phylum level (Figure 2). The treatment effects can be visually observed on the five phyla including *Actinobacteria*, *Proteobactia*, *Chloroflexi*, *Acidobacteriota*, and *Fermicutes*. For example, the abundance of *Actinobacteria* in CRU+FA was considerably higher than in CRU treatment on day 1 but the difference diminished on the following days. In contract, the abundance of *Proteobacteria* was similar between CRU and CRU+FA treatments on day 1 but that for CRU+FA became higher than in CRU on the following days.

### 3.2. Microbial Taxa Associated with CRU Combined with FA Treatment on N Conversion

Annotated potential taxa linked with N conversion by arranging bacterial OTUs into various ecological clusters (or modules, Mod), which are made up of closely related bacterial taxa (Figure 3A). Not surprisingly, distinct ecological clusters dominated the two different treatments (CRU and CRU+FA) on separate fertilizer days (Figure 3B). Changes in the available nutrients during the different treatments may result in changes in the microbial community in long-term nutrient release.

### 3.3. Application of Fulvic Acid Improved Soil Fertilizer Preserving

The CRU met the increased N need for wheat at the period of highest N use, and use efficiency; it contributes to a “fertilizer N move” in high-yield wheat fields (Appendix A). In the soil column, the leaching rate and weight moisture content of soil were significantly different between the addition of FA and the control treatments (Figure 4). In the 20 cm soil depth column, the leaching rate of CRU+FA treatment was significantly decreased by 71.8%, compared with the CRU treatment (Figure 4A). Meanwhile, the CRU+FA treatment raised the weight moisture content of soil by 54.2% compared with CRU treatment (Figure 4B).

Controlled-release urea and FA could adjust the conversion of N in 0–20 cm depths of soil column (Figure 5). The change of NO_3_^−^-N and NH_4_^+^-N content obviously increase in the treatments of CRU and CRU+FA throughout 0–60 days. Leaching NO_3_^−^-N of CRU+FA treatment was considerably decreased by 84.2% and 12.7%, respectively, contrasted to CRU treatment at 7 and 30 days (Figure 5A). Meanwhile, leaching NH_4_^+^-N of CRU+FA treatment was remarkably higher, with 87.4% on average, than for the CRU treatment (Figure 5B).

Additional correlation Heatmap analysis showed that the environmental factors of leaching solution, incubator cultivated experiment, and pot experiment were significantly linked with Mod 3 genera (Figure 6), and the genera of CRU+FA treatment were significantly changed than CRU treatment in Manhattan plot analysis (Figure 7). During different fertilizer days, Manhattan plots showing OTUs with significant differences in relative abundance in CRU and CRU+FA treatments in Mod 3 (Appendix A).

On the Mod3–1d and Mod3–7d with low N levels, the most dominant bacterial genera across all samples were *Sphingomonas*. The relative abundance of *Sphingomonas* communities in CRU+FA treatment were 14.66% and 26.09% lower, respectively than in the CRU treatment (Figure 8A,B). *Nitrospira*, *Nocardioides*, and *Ensifer* were significantly different between CRU+FA and CRU treatments, though their relative abundances were relatively lower. On Mod3–30d and Mod3–60d with high soil N levels, the most dominant bacterial community across all samples were *Lysobacter*, *Gaiella*, and *Blastococcus*. On Mod3–30d, the relative abundances of *Lysobacter* and *Gaiella* of CRU+FA treatment were 81.67% and 35.33% lower, respectively than in CRU treatment (Figure 8C,D). The relative abundance of *Blastococcus* was 13.10% higher in CRU+FA than in CRU. On Mod3–60d, the relative abundance of *Lysobacter* of CRU+FA treatment was 25.73% higher than in CRU treatment, and the relative abundance of *Nocardioides* was 43.46% less in CRU+FA than in CRU. The relative abundance of *Blastococcus* of CRU+FA treatment was 21.74% higher than in CRU treatment.

### 3.4. Environmental Drivers of Community Composition

We correlated distance-corrected dissimilarities of ammonia oxidation bacteria, nitrifying bacteria and denitrifying bacteria community composition with those of environmental factors to test whether adding FA could affect the bacterial community with ammonia oxidation process, nitrifying process and denitrifying process and to identify environmental drivers in our dataset (Figure 9). Overall, NO_3_^−^-N content, NH_4_^+^-N content, pH, NUE and yield of the pot experiment, and leaching NO_3_^−^-N content of the soil column leaching experiment were the strongest correlates of ammonia oxidation bacteria, nitrifying bacteria, and denitrifying bacteria community composition in the surface layer (Figure 9). Meanwhile, the NH_4_^+^-N content of the conversion of the soil column leaching experiment and the NO_3_^−^-N content of the conversion of N in the soil microbial incubation experiment were the strongest correlates of nitrifying bacteria community composition.

### 3.5. Wheat Yield and Fertilizer Use Efficiency Affected by CRU Combined with FA

Compared to CRU treatment, the wheat yield of CRU+FA treatment was significantly improved by 22.1% (Table 1). The CRU+FA treatment could significantly increase the wheat yield, achieving the result of synergistic interactions. Compared with CRU treatment, wheat aboveground biomass and 1000-grain weight of CRU+FA treatment were markedly raised by 16.2% and 25.9%, respectively. The effectiveness of fertilizer application is strongly dependent on crop absorption capacity and the soil’s ability to deliver nutrients (Table 2). The CRU+FA treatment increased NUE, NAE, and NPE by 41.2%, 40.9%, and 22.0% respectively, compared with CRU treatment. This shows that the application of CRU+FA could further increase NUE. 

## 4. Discussion

### 4.1. Wheat Nitrogen Cycle Process Affected by CRU Combined with FA

The coated CRU could be directly blended with FA in a one-time fertilizer application, which offers the advantage of reduced cost when compared to regular urea. Using CRU+FA does not require special supporting machinery, and the two products may result in complementary advantages and synergies. The physiology and molecular mechanisms within crops using FA as a nutrient source were that FA improved water preservation and fertilizer preservation (Figure 4 and Figure 5; Appendix A), and the CRU+FA could further increase fertilizer preservation. Furthermore, the nutrients of CRU were slowly released and the addition of macromolecular network FA could complex part of the nutrients, which further increased the time of nutrient retention in the soil. The existence of many unsaturated bonds in FA could effectively prevent the oxidation of mercapto (-SH) in urease and inhibit urease activity. The FA has high cation exchange capacity and strong adsorption capacity, which plays an important role in regulating soil nitrogen availability, keeping NH_4_^+^ and reducing ammonia volatilization.

### 4.2. The Soil Bacterial Communities Affected by CRU Combined with FA

Fulvic acid does not react with urea to produce new functional groups (Appendix A). In the indoor constant temperature incubator cultivated experiment, adding FA treatment considerably raised the relative abundance of bacteria compared to CRU alone (Figure 2). When CRU applied to the soil during the period from 1d to 7d, the early nutrients of CRU were slowly released and the nutrient concentration in the soil was low, belonging to the low nitrogen period. In the 30d to 60d period, the nutrients of CRU were released in large quantities and the nutrient concentration during the period belonged to the high nitrogen (Appendix A). The relative abundance of bacteria on different fertilizer days was significantly different due to the application of FA and varying N concentration in the soil (Figure 2B).

We discovered that N fertilizer boosted the relative abundance of *Actinobacteria* and *Proteobacteria* (Figure 2). The positive function of *Actinobacteria* and *Proteobacteria* in soils was predicted at the phylum level, and these species have been documented as having a considerable influence on the N cycle [27]. Meanwhile, soil N availability regulates microbial communities, which have significant effects on global N cycling [28]. The positive function of bacteria community of adding FA treatment was significantly increased in the low nitrogen period and explained why CRU+FA increased the yield. When urea was applied to the soil, the nutrients were quickly released, the soil microenvironment reached a high N level. Because the application of FA to high-N-level soil significantly decreased the relative abundance of beneficial of bacteria community in the soil, the nutrients were decomposed and utilized in large quantities so that the plants could not be absorbed and used, and the yield of wheat was significantly reduced.

### 4.3. Microbial Co-Occurrence Network Affected by CRU Combined with FA

After fertilization, the soil bacterial community structure of CRU+FA treatment was significantly different from that of CRU treatment on the two main coordinates (PC1 = 9.42% and PC2 = 5.54%) (Figure 1A; Appendix A). The difference in the bacterial community in CRU+FA treatment and CRU treatment was more significant as culture time increased (0–60 days) (Figure 1B; Appendix A). CRU’s slow release of nutrients is the main driving factor for the community difference, followed by the difference caused by the addition of FA to improve the soil structure [17]. The CRU and CRU+FA treatments ecological cluster of Mod 3 demonstrated that the relative abundance of bacterial communities was more intense on various days, 0.02–0.05 and 0.02–0.04, respectively (Appendix A). According to the network analysis, the bacterial community was more sensitive to CRU+FA treatment, because the network modules of bacteria treated with CRU+FA were significantly more than those of CRU (Appendix A). Species vulnerable to N fertilizer were clustered together, indicating that exogenous perturbations may have a significantly greater effect on the bacterial community by targeting vulnerable species (Appendix A). 

The nutrient release of CRU reduced the variety of interaction linkages, which is substantiated by fewer bacterial node connections (Appendix A). Moreover, the CRU+FA network expanded in number of linkages, indicating that adding FA may benefit a variety of mutualistic bacteria (Appendix A). In brief, high-N fertilizer may impact community structure stability by increasing mutualistic relationships among bacteria [3]. The N nutrient release of CRU was consistent with the nutrient absorption of wheat in soils and sophisticated ecological linkages between bacteria may protect the integrity of the community structure. The addition of FA, on the other hand, disrupted the original relationship of bacteria, stimulated the growth of dominant bacteria, altered the initial ecological network linkages, and had a synergistic impact.

### 4.4. The Soil Bacterial Community of Nitrogen Cycle Affected by CRU Combined with FA

In classic N conversion models, ammonia N is often eliminated by the bacterial nitrification and denitrification processes [29] (Yun et al., 2019), and rarely consideration has been paid to the bacteria capable of ammonia assimilation. Bacteria ammonia assimilation may straightly assimilate ammonia N and turn it into bacteria biological components [30], theoretically, the increase in ammonia assimilation will weaken the fertilizer effect of N fertilizer. Ying [31] recorded that the ammonia-oxidizing activity of bacteria is dominant in the process of the soil N cycle in neutral and alkaline soils. Under the experimental conditions, the pH of FA particles is acidic, which will affect the pH under soil microdomain conditions, significantly affect the growth and functional vitality of ammonia-oxidizing bacteria, and reduce the relative abundance of ammonia-oxidizing bacteria. Meanwhile, our results showed that adding FA treatment decreased the relative abundances of *Sphingomonas* at 1 d and 7 d, and increased soil fertility which increased the content of available N, NUE, and wheat yield, compared to CRU treatment (Figure 8A,B). The FA markedly increased the growth and efficiency of nitrogen fixation [32].

It has been reported that many bacterial communities in *Proteobacteria* are involved in soil nitrogen cycle, and most of the bacteria community related to nitrification belong to *Proteobacteria* [33]. In this study, N application expressively promoted the relative abundance of *Proteobacteria*, and the relative abundance of *Lysobacter* and *Nitrospira* increased with the increase in N concentration. It revealed that the relative abundance of *Nitrospira* was dominant at low N concentration, and *Lysobacter* increased significantly under high N concentration. Most nitrifying bacteria belong to aerobic autotrophic microorganisms, which need to complete oxidization of NH_4_^+^ to produce ATP with the participation of O_2_ [34]. There is a negative correlation between oxygen partial pressure and water content. Excessively low and excessively high water content have adverse effects on soil nitrification [35]. The addition of FA significantly increased soil water content. When the micro-domain water content was high, the oxygen content in soil was low to inhibit the metabolism of nitrifying microorganisms [36]. On the 30th day, the addition of FA could significantly reduce the relative abundance of *Lysobacter*, reduce the transformation of NH_4_^+^-N, and indirectly promote the absorption of NH_4_^+^-N by wheat roots (Figure 8C). However, with the continuous high N concentration level for a long time (60th day), the addition of FA will also significantly increase the relative abundance of *Lysobacter*, promote the transformation of NH_4_^+^-N, and reduce the NUE.

There were two dominant genera, *Nocardioides* and *Gaiella*, belonging to *Actinobacteria*, which could reduce nitrate to nitrite in soil [37]. The dominant bacteria communities were different with different N concentrations in soil. Under the condition of low N concentration, the relative abundance of *Nocardioides* was dominant. With the increase in soil N concentration, the addition of FA could dramatically inhibit the relative abundance of *Nocardioides*, decrease the transformation of NO_3_^−^-N, and promote the absorption of wheat roots [38]. On the 30th day, *Gaiella* was the main dominant bacterium. Adding FA could significantly reduce the abundance of *Gaiella* and effectively stabilize the concentration of NO_3_^−^-N in soil (Figure 8C). The FA contains many carboxyl and phenolic groups, which affect the proton transfer and the balance between gaseous nitrite and nitrite ions (NO_2_^−^). The carboxyl and phenolic groups in FA could combine with the generated NO_2_^−^ through covalent bond to reduce the substrate concentration in the denitrification process [39], and then affect the growth and reproduction of denitrifying bacteria to inhibit the denitrification process, increase the NUE, and weaken environmental pollution [40].

*Ensifer*, *Blastococcus*, and *Pseudolabrys* all play a role in soil nitrogen fixation capacity [41,42], and their relative abundances were significantly increased. The FA contains many active ions, which could promote the growth and nitrogen fixation ability of *Ensifer*, *Blastococcus*, and *Pseudolabrys* with organic carbon as a carbon source. *Ensifer*, *Blastococcus*, and *Pseudolabrys* could fix inorganic N in soil, and could be biologically fixed to microorganisms in the process of growth and reproduction. Meanwhile, *Ensifer*, *Blastococcus*, and *Pseudolabrys* proliferation and expansion could alter the dominant population of soil microorganisms and limit the growth of some alien microorganisms. Wang [27] and Zhou [28] studied the effect of N fertilizer on soil microbial biomass and inorganic N, and determined that N fertilizer significantly inhibited the abundance of soil microorganisms. In the experiment, the content of inorganic N treated with FA was significantly increased. The reason was that the rapid growth and reproduction in *Ensifer*, *Blastococcus*, and *Pseudolabrys* increased the NH_4_^+^-N fixation to microbial nitrogen and reduced the nitrification and denitrification process [43,44,45]. However, when treated with CRU+FA, the inorganic N content was significantly higher than that treated with CRU. This difference may occur due to the fact that the application of FA could provide a carbon source for the growth of microorganisms, while the nutrients release of CRU were slow and required a certain amount of heat and water. Thus, *Ensifer*, *Blastococcus*, and *Pseudolabrys* were unable to directly consume all the inorganic nitrogen of the CRU and fix it to microbial nitrogen. Most of the bacterial communities considerably affected by adding FA treatment had a favorable influence on wheat growth promotion, yield, and NUE.

## 5. Conclusions

The CRU+FA treatment significantly reduced the relative abundance of bacteria involved in nitrification and denitrification reactions, while increasing the relative abundance of bacteria involved in N fixation reduced NO_3_^−^-N leaching by 12.7–84.2% and increased wheat yield by 22.1% when compared to the CRU treatment. In summary, the combined effects of slow release of nutrients provided by CRU and promotion growth of FA improved the soil microbial communities of nitrogen transformation, reduced nutrition loss, and increased wheat yield.

## Figures and Tables

**Figure 1 microorganisms-10-01823-f001:**
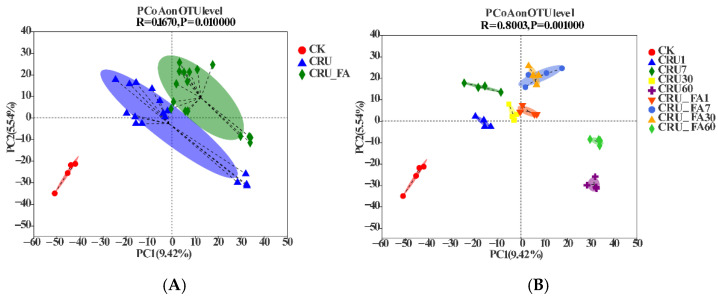
Principal coordinate analysis (PCoA) based on the weighted UniFrac distance matrix from different treatments (**A**) and different incubation days (**B**).

**Figure 2 microorganisms-10-01823-f002:**
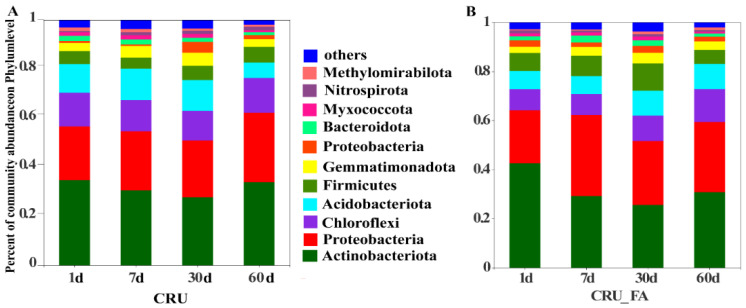
The treatment effects on relative abundance of soil bacteria community at the phylum level for CRU (**A**) and CRU+FA (**B**). Values reported are the means of three replicates.

**Figure 3 microorganisms-10-01823-f003:**
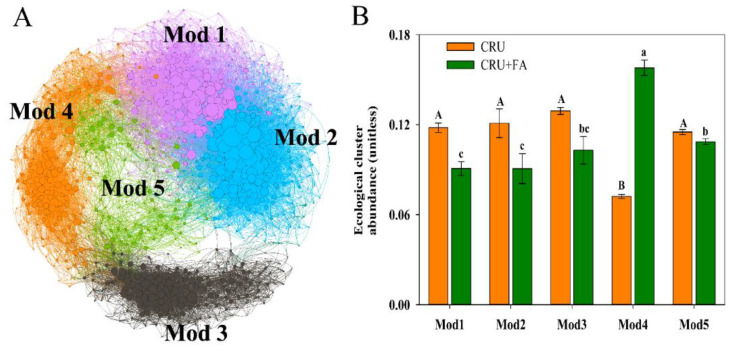
Microbial network diagram with nodes (taxa) colored by each of the major five ecological clusters (modules, Mod) of bacterial communities (**A**); the relative abundance of the five modules (**B**). Capital letters indicate significant differences between the CRU treatment on different modules (*p* < 0.05). Lowercase letters indicate significant differences between the CRU+FA treatment on different modules (*p* < 0.05).

**Figure 4 microorganisms-10-01823-f004:**
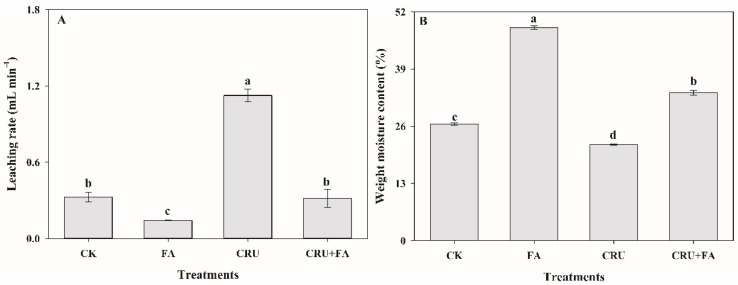
The leaching rate (**A**) and weight moisture content (**B**) after adding FA. Treatments or treatment components: CK (Control no added fertilizer); FA (fulvic acid 90 kg ha^−1^); CRU (N from controlled-release urea 450 kg ha^−1^); CRU+FA (N from controlled-release urea 450 kg ha^−1^, fulvic acid 90 kg ha^−1^). Bar heights represent the averages of three replicates and error bars represent ± SE. Within each graph, means followed with the same letter were not significantly different based on a one-way ANOVA followed by Duncan’s multiple-range test (*p* > 0.05).

**Figure 5 microorganisms-10-01823-f005:**
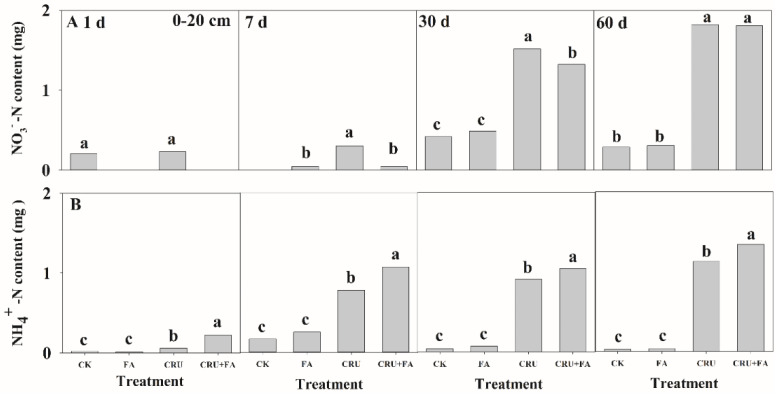
The nutrient change: NH_4_^+^-N content (ammonium nitrogen content, (**A**) and NO_3_^−^-N content (nitrate nitrogen content, (**B**) after adding FA from leaching solution. Treatments or treatment components: CK (Control no added fertilizer); FA (fulvic acid 90 kg ha^−1^); CRU (N from controlled-release urea 450 kg ha^−1^); CRU+FA (N from controlled-release urea 450 kg ha^−1^, fulvic acid 90 kg ha^−1^). Within each graph, means followed with the same letter were not significantly different based on a one-way ANOVA followed by Duncan’s multiple-range test (*p* > 0.05).

**Figure 6 microorganisms-10-01823-f006:**
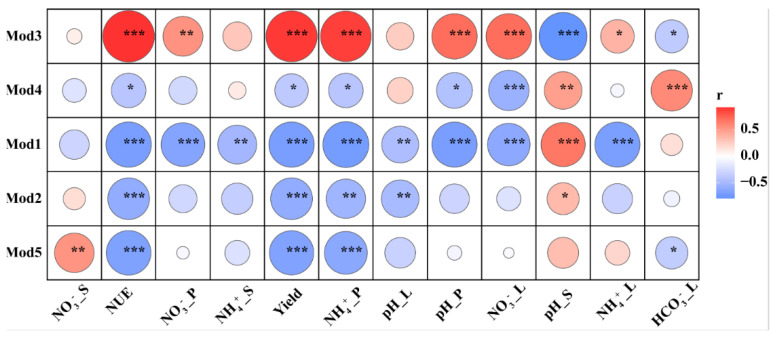
Pearson correlation heat maps of environmental factors and the five modules. Red represents a positive correlation, and blue represents a negative correlation (* *p* < 0.05; ** *p* < 0.01; *** *p* < 0.001). NO_3_^−^_L: the nitrate nitrogen from leaching solution; NH_4_+_L: the ammonium nitrogen from leaching solution; HO_3_^−^_L: the bicarbonate from leaching solution; pH_L: the pH value from leaching solution; NO_3_^−^_S: the nitrate nitrogen from incubator cultivated experiment; NH_4_^+^_S: the ammonium nitrogen from incubator cultivated experiment; pH_S: the pH value from incubator cultivated experiment; NO_3_^−^_P: the nitrate nitrogen from pot experiment; NH_4_^+^_P: the ammonium nitrogen from pot experiment; pH_P: the pH value from pot experiment; NUE: N use efficiencies of wheat from pot experiment; Yield: the grain of wheat from pot experiment.

**Figure 7 microorganisms-10-01823-f007:**
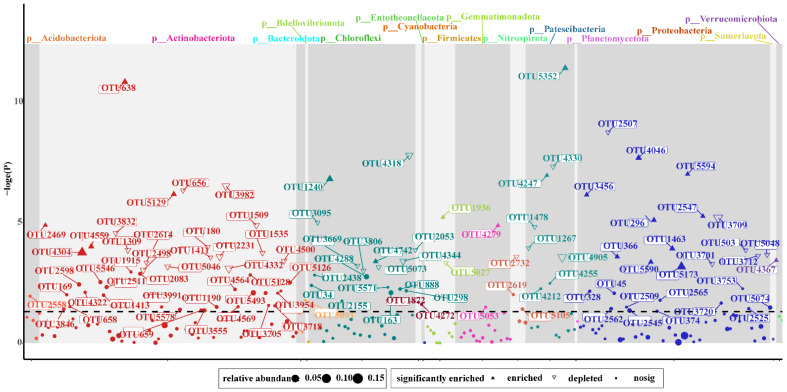
Manhattan plots showing OTUs with significant differences in relative abundance in CRU and CRU+FA treatments in Mod 3. Each triangle or circle represents an individual OTU. Upward solid triangles or downward hollow triangles represent OTUs enriched or depleted, respectively, whereas circles represent OTUs that are not significantly enriched or depleted. The dashed line corresponds to the false discovery rate-corrected threshold *p*-value for significance (α = 0.05). The color of each dot represents the taxonomic affiliation of the OTU (phylum level) and the size corresponds to its relative abundance in the samples.

**Figure 8 microorganisms-10-01823-f008:**
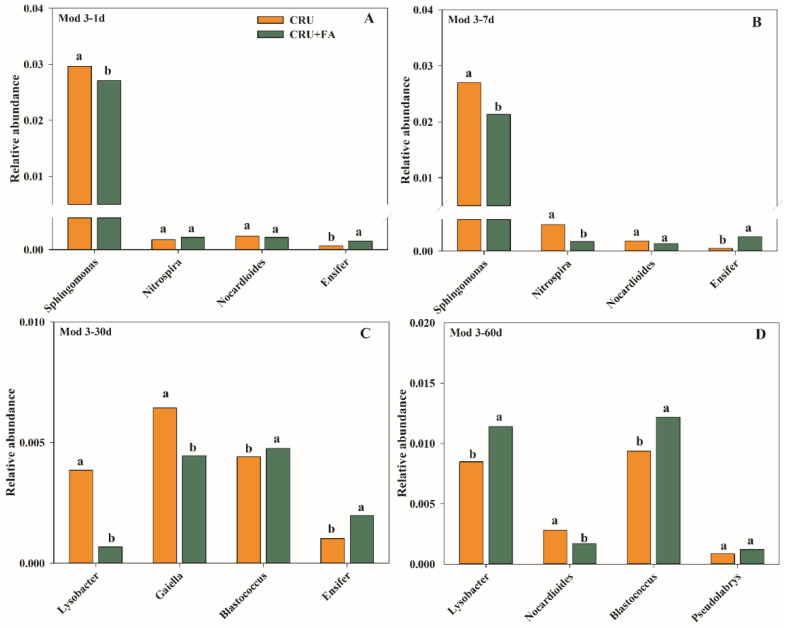
Significant difference potential genera of Mod 3 associated with nitrogen cycle in different incubation days: 1 d (**A**), 7 d (**B**), 30 d (**C**) and 60 d (**D**). Within each graph, means followed with the same letter were not significantly different based on a one-way ANOVA followed by Duncan’s multiple-range test (*p* > 0.05).

**Figure 9 microorganisms-10-01823-f009:**
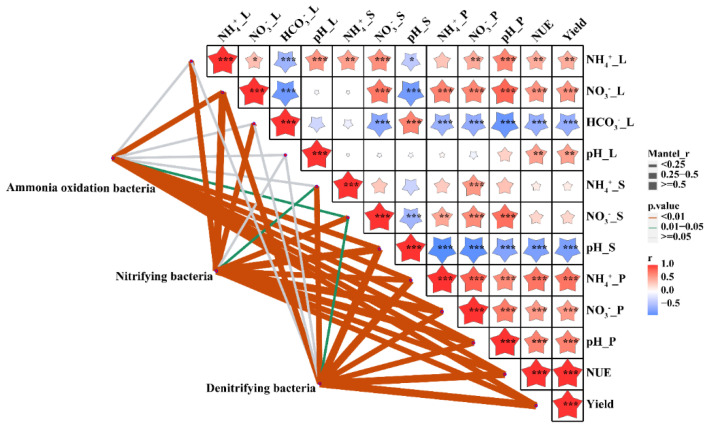
Pairwise comparisons of environmental factors are shown, with a color gradient denoting Spearman’s correlation coefficient. Taxonomy is based on two independent methods: diversity and OTUs of difference potential genera was related to each environmental factor by partial (geographic distance corrected) Mantel tests. Edge width corresponds to Mantel’s r statistic for the corresponding distance correlations, and edge color denotes the statistical significance based on 9999 permutations. “*” represents a correlation between the two indicators, “**” means a significant correlation between the two indicators, and “***” means a very significant correlation between the two indicators. NO_3_^−^_L: the nitrate nitrogen from leaching solution; NH_4_^+^_L: the ammonium nitrogen from leaching solution; HO_3_^−^_L: the bicarbonate from leaching solution; pH_L: the pH value from leaching solution; NO_3_^−^_S: the nitrate nitrogen from incubator cultivated experiment; NH_4_^+^_S: the ammonium nitrogen from incubator cultivated experiment; pH_S: the pH value from incubator cultivated experiment; NO_3_^−^_P: the nitrate nitrogen from pot experiment; NH_4_^+^_P: the ammonium nitrogen from pot experiment; pH_P: the pH value from pot experiment; NUE: N use efficiencies of wheat from pot experiment Yield: the grain of wheat from pot experiment.

**Table 1 microorganisms-10-01823-t001:** Yield and yield components of wheat with different treatments.

Treatment ^a^	Grain Yield (g pot^−1^) ^b^	Aboveground Biomass (g pot^−1^) ^b^	1000-Grain Weight (g) ^b^	Seeds per Spike ^b^	Spike No. per Plant ^b^	Yield Increase (%) ^c^
Control	37.6 ^c^	82.6 ^c^	53.6 ^b^	21.7 ^b^	39.0 ^b^	−53.9
CRU	81.6 ^b^	168.8 ^b^	47.1 ^c^	30.2 ^a^	71.5 ^a^	-
CRU+FA	99.6 ^a^	196.2 ^a^	59.3 ^a^	32.8 ^a^	65.0 ^a^	22.1 **

^a^ Treatments: No N (control); controlled-release urea (CRU); controlled-release urea combined with FA (CRU+FA). ^b^ Means within each column followed by the same letters were not significantly different based on a one-way ANOVA followed by Duncan ’s multiple-range test (*p* > 0.05). ^c^ Compared with controlled-release urea. “**” means 0.001 < *p* < 0.01.

**Table 2 microorganisms-10-01823-t002:** Fertilizer use efficiency of wheat under with different treatments.

Treatment ^a^	NUE (%) ^b^	Change (%) ^c^	NAE (kg kg^−1^) ^b^	NPE (kg kg^−1^) ^b^
Control	-	-	-	-
CRU	35.4 ^b^	-	7.0 ^b^	13. 0 ^b^
CRU+FA	50.0 ^a^	41.2 *	9.9 ^a^	15.9 ^a^

^a^ Treatments: No N (control); controlled-release urea (CRU); controlled-release urea combined with FA (CRU+FA). ^b^ Means within each column followed by the same letters were not significantly different based on a one-way ANOVA followed by Duncan ’s multiple-range test (*p* > 0.05) ^c^ Compared with controlled-release urea. NUE: N use efficiency; NAE: N agronomic efficiency; NPE: N physiological efficiency. “*” means 0.01 < *p* < 0.05.

## Data Availability

The raw sequence reads of 16S rRNA gene amplicons have been deposited to the National Center for Biotechnology Information (NCBI)Sequence Read Archive (SRA), under BioProject ID PRJNA857168.

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
