# Peer review of "Soil Microbial Co-Occurrence Patterns under Controlled-Release Urea and Fulvic Acid Applications"

_microorganisms, 2022, doi:10.3390/microorganisms10091823_

Round 1

Reviewer 1 Report

The authors have written a well-structured article. The authors did a lot of work of experimental design, data analysis, and the results presenting.

I just have a few some suggestions:

1) Please add in the Introduction or Materials and Methods why you are focusing on these soil microorganisms.

2) Fig. 7 and Supplementary Fig. S4 are not clear. Whether it is possible to improve their quality in some way?

I have no further comments. In my opinion, after minor corrections the article is suitable for publication.

Author Response

The authors have written a well-structured article. The authors did a lot of work of experimental design, data analysis, and the results presenting.

Response: Thank you for accepting our research, and we will carefully enhance the manuscript in response to your suggestions.

I just have a few some suggestions:

Point 1: Please add in the Introduction or Materials and Methods why you are focusing on these soil microorganisms.

Response 1: Thanks for your comment. We are very sorry for our unclearly expressing. We have revised the sentences in the Introduction.

“Microorganisms are the engine that drives the biogeochemical cycle of soil elements. N addition directly affects the community composition, activity, and metabolic rate of microorganisms by changing the availability of soil N, which then affects soil organic matter transformation and the carbon and N cycle of the ecosystem. The soil microbial communities are an important factor affecting N cycling and U use efficiency, however, the impact of CRU combined with FA on the soil microbial communities has not been reported.” (Lines 73-79)

Point 2: Fig. 7 and Supplementary Fig. S4 are not clear. Whether it is possible to improve their quality in some way?

Response 2: Thank you for your advice for the paper. We have improved the quality of Fig. 7 and Supplementary Fig. S6 (original figure was Supplementary Fig. S4).

Fig. 7 Manhattan plots showing OTUs with significant differences in relative abundance in CRU and CRU+FA treatments in Mod 3. Each triangle or circle represents an individual OTU. Upward solid triangles or downward hollow triangles represent OTUs enriched or depleted, respectively, whereas circles represent OTUs that are not significantly enriched or depleted. The dashed line corresponds to the false discovery rate-corrected threshold P-value for significance (α = 0.05). The color of each dot represents the taxonomic affiliation of the OTU (phylum level), and the size corresponds to its relative abundance in the samples. (Lines 309-314)

Supplementary Fig. S6 Manhattan plots showing OTUs with significant differences in relative abundance in CRU and CRU+FA treatments in Mod 3 during different fertilizer days. Each triangle or circle represents an individual OTU. Upward solid triangles or downward hollow triangles represent OTUs enriched or depleted, respectively, whereas circles represent OTUs that are not significantly enriched or depleted. The dashed line corresponds to the false discovery rate-corrected threshold P-value for significance (α = 0.05). The color of each dot represents the taxonomic affiliation of the OTU (phylum level), and the size corresponds to its relative abundance in the samples.

Reviewer 2 Report

See attached file

Author Response

Point 1: Keywords: controlled-release urea, fulvic acid, soil bacterial communities, nutrition change, ecological network. Please change some keywords. Title and key words must not contain the same words.

Response 1: Thanks for your comment. We have changed the keywords to distinguish it from the title.

“Keywords: nitrogen, synergist, soil bacterial community, nutrition change, ecological network”. (Lines 34-35)

Point 2: Line 75. What was not examined is how CRU combined with FA would change soil microbial community, which is a key factor affecting N cycling and U use efficiency. I suggest to rewrite this sentence.

Response 2: We are very sorry for our expression, and thank you for your comment. We have rewritten the sentence.

“The soil microbial communities are an important factor affecting N cycling and U use efficiency, however, the impact of CRU combined with FA on the soil microbial communities has not been reported.” (Lines 77-79)

Point 3: Line 77. Given the slow and controlled N releasing pattern of CRU for improve soil fertility environment [18, 19] and FA’s ability to change soil properties and promote the growth and multiplication of beneficial soil microbes [20, 21]. What….?

Response 3: We are very sorry for our unclearly expressing. We have rewritten the sentence.

“Given the nutrients slow-release of CRU can enhance the soil fertility environment [18, 19], FA modifies soil characteristics, boost crop root growth, and increases the multiplication of beneficial soil microorganisms [20, 21].” (Lines 82-86)

Point 4: Line 89. Please provide the soil type.

Response 4: Thanks for your comment. We have added the soil type.

“The soil sample was obtained at the top 20 cm depth from a research farm (36°9′40′′N, 117°9′48′′E) in Shandong Province, China; which was classified into the Typic Hapli-Udic Argosol in Chinese soil classification.” (Lines 95-97)

Point 5: Line 135. Soil examples sampled from 0-20 cm depth on days 33, 148, 174, and 204 after fertilization for analyses of soil pH, nitrate N, and ammonia N contents (Supplementary Table S1). Please explain the reasons why those days are chosen and not others.

Response 5: Thanks for your comment. We have explained the reasons.

The seedling stage, re-greening stage, jointing stage, and booting stage of wheat occur 33, 148, 174, and 204 days after fertilization, respectively, and are critical stages of wheat nutrient absorption. Therefore, we chose these time points for sampling.

“Soil examples sampled from 0-20 cm depth on days 33 (seedling stage), 148 (re-greening stage), 174 (jointing stage), and 204 (booting stage) after fertilization for analyses of soil pH, nitrate N, and ammonia N contents (Supplementary Table S1).” (Lines 146-148)

Point 6: Line 144. A pH meter (PB-10, Germany) was used to evaluate…To evaluate or to determine?

Response 6: We are very sorry for our unclearly expressing. We have replaced “evaluate” with “determine”.

“A pH meter (PB-10, Germany) was used to determine the pH of the soil and fulvic acid using water extraction without CO2 (soil: water was 1:2.5).” (Lines 155-156)

Point 7: Please improve the figure 1

Response 7: Thanks for your comment. We have modified the figure 1.

Fig. 1 Principal coordinate analysis (PCA) based on the weighted UniFrac distance matrix from different treatments (A) and different incubation days (B). (Lines 237-238)

Point 8: Please rewrite the conclusion section

Response 8: Thanks for your comment. We have rewritten the Conclusion.

“The CRU+FA treatment significantly reduced the relative abundance of bacteria involved in nitrification and denitrification reactions, while increasing the relative abundance of bacteria involved in N fixation, reducing NO3--N leaching by 12.7%-84.2% and increasing wheat yield by 22.1% when compared to the CRU treatment. In summary, the combined effects of nutrients slow-release of CRU and promotion growth of FA improved the soil microbial communities of nitrogen transformation, reduced nutrition loss, and increased wheat yield.” (Lines 519-525)

Point 9: I suggest to include some experimental photos.

Response 9: Thanks for your comment. We have added some experimental photos.

Supplementary Fig. S2 The experimental device of soil Column Leaching Experiment. (Lines 124-126)

Supplementary Fig. S3 The experimental process of pot experiment. (Lines 138-140)

Reviewer 3 Report

Line 17. Should read "were carried out" instead of "were carried"

Line 23. Delete "across of"

Lines 26–30. This sentence is much too long. Split it up into two separate thoughts.

Line 32. Replace "nutrition losing"by "loss of nutrients"

Line 38. Replace "food enough" by "enough food"

Line 43. Delete "the"

Line 46. Delete "The"

Line 48. Replace "unitary" by "single"

Lines 58–59. It is probably better to replace "Fulvic acid, on the other hand," by "However, fulvic acid"

Line 66. Replace "the growth and yield of crop" by "crop growth and yield"

Lines 71–75. Split this long sentence up into two sentences by putting a period (full stop) after "soil elements".

Lines 77–79. This is a sentence fragment, not a sentence. It needs rewriting.

Line 84. Replace "related" by "relate"

Line 101. Section 2.2. From lines 102–104, it seems that the experimental design is a one-way classification, i.e. 3 treatments each with 3 replicates. If so, this is a very small experiment, giving only 6 degrees of freedom for the error term, which may make it difficult to reject the null hypothesis. Is my interpretation correct? The same can be said for the experiments described in Sections 2.3 and 2.4.

Line 109. "and each sampling time was taken as three repetitions of each treatment" needs rewriting. It doesn't read well as it stands.

Line 113. There are 4 treatments, but no results are given in Table S1 for the FA treatment or for the control. Why is that?

Line 146. "was derivative"? What does this mean? Needs rewriting.

Line 179. Should read "data were" rather than "data was"

Line 201. Should that read "calculated using the R environment"?

Line 203. Are there really "two distinct methods" here?

Line 230. Replace "upper" by "higher"

Line 260. After "Control", add the symbol "CK"

Line 261. After the means the averages of three replicates? If so, that should be stated.

Line 263. "test", not "tests". The same applies to line 275 and line 315.

Line 265. Delete "of" after "content"

Line 269. "on average", not "averagely"

Line 303. "lower" rather than "lowed"

Line 308. Replace "more" by "higher". Also, the same for lines 309 and 311.

Line 330. Replace "Taxonomic" by "Taxonomy is"

Lines 344–345. Replace "remarkably" by "markedly"

Line 352. Replace "analysis of variance" by the name of the pairwise test that was used (was it Duncan's?)

Line 356. As above, was it Duncan's?

Line 361. Delete comma after "coated"

Line 376. Replace "cultivating" by "cultivated experiment"

Line 391. Replace "explain the" by "explained why"

Line 380. Instead of "the 30d to 60d", it would be better to have "in the 30d to 60d period"

Line 383. "different" rather than "difference"

Line 386 and in several other places. Proteobacteria should consistently be in italics.

Line 391. Insert "was" between "urea" and "applied"

Line 394. "abundance of advantage of bacteria" doesn't make sense. Needs rewriting.

Sections 4.3 and 4.4. Extensive rewriting is needed in these sections. For example, lines 469–470, "The abundance of Ensifer, Blastococcus and Pseudolabrys all play a role in soil nitrogen fixation capacity [41, 42] were significantly increased" doesn't make sense as it stands.

Line 487. "fix", not "fixe"

Line 503. The degree symbol is missing.

Supplementary Table S1. Are the entries the means of 3 replicates? It should be stated what these are. Can standard errors be added?

Author Response

Point 1: Line 17. Should read "were carried out" instead of "were carried"

Response 1: Thanks for your comment. We have replaced " were carried " with " were carried out ".

“Herein, a 0-60 days laboratory experiment and a consecutive pot experiment (2016-2018) were carried out to reveal the effects of using CRU on soil microbial N-cycling processes and soil fertility, with and without FA application.” (Lines 16-18)

Point 2: Line 23. Delete "across of".

Response 2: Thank you for your careful modification. We have deleted "across of".

“The most dominant bacterial phyla Actinobacteria and Proteobacteria were increased with CRU+FA treatment during 0-60 days.” (Lines 23-24)

Point 3: Lines 26–30. This sentence is much too long. Split it up into two separate thoughts.

Response 3: Thanks for your comment. We have divided this sentence into two sentences.

“The CRU+FA treatment, in particular, significantly decreased the relative abundance of Sphingomonas, Lysobacter and Nitrospira associated with nitrification reactions, Nocardioides and Gaiella related to denitrification reactions. Meanwhile, the CRU+FA treatment grew the relative abundance of Ensifer, Blastococcus and Pseudolabrys that function in N fixation, and then could reduce NH4+-N and NO3--N leaching and improve the soil nutrient supply.” (Lines 26-31)

Point 4: Line 32. Replace "nutrition losing" by "loss of nutrients".

Response 4: Thanks for your comment. We have replaced "nutrition losing" with "loss of nutrients".

“In conclusion, the synergistic effects of slow nutrition release of CRU and growth promoting of FA could improve the soil microbial community of N cycle, reduce the loss of nutrients and increase the wheat yield.” (Lines 31-33)

Point 5: Line 38. Replace "food enough" by "enough food".

Response 5: Thanks for your comment. We have replaced "food enough" with "enough food".

“Nitrogen (N) is the essential nutrient for plants, and a massive amount of N fertilizers is applied to the soil to maintain food enough food in the world.” (Lines 38-39)

Point 6: Line 43. Delete "the".

Response 6: Thank you for your careful modification. We have deleted "the".

“To this end, controlled-release urea (CRU) has been developed as an alternative for the conventional urea while fulvic acids (FA) have been applied as a fertilizer syner-gist to enhance N use efficiency [3].” (Lines 44-46)

Point 7: Line 46. Delete "The".

Response 7: Thanks for your comment. We have deleted "The".

“CRU releases N at a pace that nearly matches crop nutrient absorption when com-pared with the conventional N chemical fertilizer such as urea [4].” (Lines 47-48)

Point 8: Line 48. Replace "unitary" by "single".

Response 8: Thanks for your comment. We have replaced "unitary" with "single".

" In addition to increasing crop yield, a single application of CRU can save time and labor compared with multiple applications of conventional N fertilizer [5]." (Lines 48-50)

Point 9: Lines 58–59. It is probably better to replace "Fulvic acid, on the other hand," by "However, fulvic acid".

Response 9: Thanks for your comment. We have replaced "Fulvic acid, on the other hand," with "However, fulvic acid".

“However, fulvic acid is made up of a variety of aliphatic and aromatic structures with several distinct functional (mostly oxygen-containing) groups that retain nutrient elements in the soil through cation exchange, chelation, complexation and adsorption [9].” (Lines 59-63)

Point 10: Line 66. Replace "the growth and yield of crop" by "crop growth and yield".

Response 10: Thanks for your comment. We have replaced "the growth and yield of crop" with "crop growth and yield".

“Other functionalities of fulvic acids include improving soil physical properties by promoting the formation of soil aggregates [10], enhancing plant physiological characteristics such as increasing photosynthesis and reducing transpiration [11, 12], in-creasing plant tolerance of environmental stresses [13], stimulating membrane stability and enzyme activity related with N metabolism [14], and ultimately increasing crop growth and yield [15, 16].” (Lines 63-68)

Point 11: Lines 71–75. Split this long sentence up into two sentences by putting a period (full stop) after "soil elements".

Response 11: Thank you for your careful modification. We have modified the sentence.

"Microorganisms are the engine that drives the biogeochemical cycle of soil elements. N addition directly affects the community composition, activity, and metabolic rate of microorganisms by changing the availability of soil N, which then affects soil organic matter transformation and the carbon and N cycle of the ecosystem." (Lines 73-79)

Point 12: Lines 77–79. This is a sentence fragment, not a sentence. It needs rewriting.

Response 12: Thanks for your comment. We have rewritten the sentence.

"Given the slow-release nutrients of CRU can enhance the soil fertility environment [18, 19], FA modifies soil properties, boost crop root growth, and increases the multiplication of beneficial soil microorganisms [20, 21]." (Lines 82-86)

Point 13: Line 84. Replace "related" by "relate".

Response 13: Thanks for your comment. We have replaced "related" with "relate".

“(3) evaluate soil microbial factors as they relate to crop yield and N uses efficiency with CRU combined with FA.” (Lines 91-92)

Point 14: Line 101. Section 2.2. From lines 102–104, it seems that the experimental design is a one-way classification, i.e. 3 treatments each with 3 replicates. If so, this is a very small experiment, giving only 6 degrees of freedom for the error term, which may make it difficult to reject the null hypothesis. Is my interpretation correct? The same can be said for the experiments described in Sections 2.3 and 2.4.

Response 14: Thanks for your comment. Our pot experiment is a wheat-maize rotation design with conventional urea treatment, controlled-release urea treatment, conventional urea combined with fulvic acid treatment, and controlled-release urea combined with fulvic acid treatment for two years. The most noticeable effect on increasing yield is found to be controlled-release urea treatment and controlled-release urea combined with fulvic acid treatment. The purpose of this study is to reveal the increasing yield mechanism of controlled-release urea and controlled-release urea combined with fulvic acid using the soil microbial incubation experiment and the soil column leaching experiment. As a result, we chose three important treatments as the research object: control, controlled-release urea, and controlled-release urea combination with fulvic acid, and each experiment met the statistical requirements. Simultaneously, the nutrient release rate of controlled-release urea was varied at different times, and adding sampling time points from different days, boosted the experiment's richness.

Point 15: Line 109. "and each sampling time was taken as three repetitions of each treatment" needs rewriting. It doesn't read well as it stands.

Response 15: We are very sorry for our expression and thank you for your comment. We have rewritten the sentence.

“Soil samples were obtained via destructive sampling on days 1, 7, 30, and 60, and each treatment was sampled to obtain three replicates at each sampling time.” (Lines 116-118)

Point 16: Line 113. There are 4 treatments, but no results are given in Table S1 for the FA treatment or for the control. Why is that?

Response 16: Thanks for your comment. The soil column leaching experiment contained the control and FA treatments, however, in order to compare the changes of soil nutrients in soil microbial incubation, the soil column leaching and the pot experiment at the same time, we chose CRU and CRU+FA treatments.

Point 17: Line 146. "was derivative"? What does this mean? Needs rewriting.

Response 17: We are very sorry for our expression and thank you for your comment. We have rewritten the sentence.

“Soil NO3--N and NH4+-N of inorganic N was extracted by 0.01 M CaCl2 (soil: water was 1:10) and determined using a succession flow injection analyzer (AA3-A001-02E, Germany).” (Lines 156-158)

Point 18: Line 179. Should read "data were" rather than "data was".

Response 18: Thanks for your comment. We have replaced "data was" with "data were".

“The data were processed in Microsoft Excel 2010, and the figures were created in SigmaPlot 2010 (Version 12.0, Systat Software Inc., USA).” (Lines 190-192)

Point 19: Line 201. Should that read "calculated using the R environment"?

Response 19: Thanks for your comment. We have modified the sentence.

“The module that was most positively correlated with environmental factors was identified and OTU association within the module was displayed with a Manhattan plot calculated using the R environment (http://www.r-project.org).” (Lines 210-212)

Point 20: Line 203. Are there really "two distinct methods" here?

Response 20: We are very sorry for our expression and thank you for your comment. We have rewritten the sentence.

“Taxonomic classification is based on the method: Mantel tests were used to determine the relationship between diversity and OTUs of different possible taxa and each environmental condition.” (Lines 214-216)

Point 21: Line 230. Replace "upper" by "higher".

Response 21: Thanks for your comment. We have replaced "upper" with "higher".

“For example, the abundance of Actinobacteria in CRU+FA was considerably higher than in CRU treatment on day 1 but the difference diminished on the following days.” (Lines 242-244)

Point 22: Line 260. After "Control", add the symbol "CK".

Response 22: Thanks for your comment. We have modified the sentence.

“Treatments or treatment components: CK (Control no added fertilizer); FA (Fulvic acid 90 kg ha-1); CRU (N from controlled-release urea 450 kg ha-1); CRU+FA (N from controlled-release urea 450 kg ha-1, Fulvic acid 90 kg ha-1).” (Lines 272-274)

“Treatments or treatment components: CK (Control no added fertilizer); FA (Fulvic acid 90 kg ha-1); CRU (N from controlled-release urea 450 kg ha-1); CRU+FA (N from controlled-release urea 450 kg ha-1, Fulvic acid 90 kg ha-1).” (Lines 287-289)

Point 23: Line 261. After the means the averages of three replicates? If so, that should be stated.

Response 23: Thanks for your comment. We have rewritten the sentence.

“Bar heights represent the averages of three replicates and error bars represent ± SE.” (Lines 274-275)

Point 24: Line 263. "test", not "tests". The same applies to line 275 and line 315.

Response 24: Thanks for your comment. We have replaced "tests" with "test".

“Within each graph, means followed with the same letter were not significantly different based on a one-way ANOVA followed by Duncan’s multiple-range test (P > 0.05).” (Lines 275-277)

“Within each graph, means followed with the same letter were not significantly different based on a one-way ANOVA followed by Duncan’s multiple-range test (P > 0.05).” (Lines 289-291)

“Within each graph, means followed with the same letter were not significantly different based on a one-way ANOVA followed by Duncan’s multiple-range test (P > 0.05).” (Lines 331-332)

Point 25: Line 265. Delete "of" after "content".

Response 25: Thanks for your comment. We have deleted "of" after "content".

“The change of NO3--N and NH4+-N content obviously increase in the treatments of CRU and CRU+FA, throughout 0-60 days.” (Lines 279-280)

Point 26: Line 269. "on average", not "averagely".

Response 26: Thanks for your comment. We have replaced "averagely" with "on average".

“Meanwhile, leaching NH4+-N of CRU+FA treatment was remarkably higher by 87.4%, on average, than for the CRU treatment (Fig. 5B).” (Lines 282-284)

Point 27: Line 303. "lower" rather than "lowed".

Response 27: Thanks for your comment. We have replaced "lowed" with "lower".

Nitrospira, Nocardioides, and Ensifer were significantly different between CRU+FA and CRU treatments, though their relative abundance were relatively lower.” (Lines 318-319)

Point 28: Line 308. Replace "more" by "higher". Also, the same for lines 309 and 311.

Response 28: Thanks for your comment. We have replaced "more" with "higher".

“The relative abundance of Blastococcus was 13.10% higher in CRU+FA than in CRU.” (Lines 323-324)

“On Mod3-60d, the relative abundance of Lysobacter of CRU+FA treatment was 25.73% higher than in CRU treatment, and the relative abundance of Nocardioides was 43.46% less in CRU+FA than in CRU.” (Lines 324-326)

“The relative abundance of Blastococcus of CRU+FA treatment were 21.74% higher than in CRU treatment.” (Lines 326-328)

Point 29: Line 330. Replace "Taxonomic" by "Taxonomy is".

Response 29: Thanks for your comment. We have replaced "Taxonomic" with "Taxonomy is".

“Taxonomic is based on two independent methods: Diversity and OTUs of difference potential genera was related to each environmental factor by partial (geographic distance corrected) Mantel tests.” (Lines 347-348)

Point 30: Lines 344–345. Replace "remarkably" by "markedly".

Response 30: Thanks for your comment. We have replaced "remarkably" with "markedly".

“Compared with CRU treatment, wheat aboveground biomass and 1000-grain weight of CRU+FA treatment were markedly raised by16.2% and 25.9%, respectively.” (Lines 360-362)

Point 31: Line 352. Replace "analysis of variance" by the name of the pairwise test that was used (was it Duncan's?).

Response 31: Thanks for your comment. We have rewritten the sentence.

“Means within each column followed by the same letters were not significantly different based on a one-way ANOVA followed by Duncan ’s multiple-range test (P > 0.05).” (Lines 369-370)

Point 32: Line 356. As above, was it Duncan's?

Response 32: Thanks for your comment. We have rewritten the sentence.

“Means within each column followed by the same letters were not significantly different based on a one-way ANOVA followed by Duncan ’s multiple-range test (P > 0.05).” (Lines 375-376)

Point 33: Line 361. Delete comma after "coated".

Response 33: Thanks for your comment. We have deleted "comma after "coated"".

“The coated CRU could be directly blended with FA in a one-time fertilizer application, which offers the advantage of reduced cost compared to regular urea.” (Lines 381-382)

Point 34: Line 376. Replace "cultivating" by "cultivated experiment".

Response 34: Thanks for your comment. We have replaced "cultivating" with "cultivated experiment".

“In the indoor constant temperature incubator cultivated experiment, adding FA treatment considerably raised relative abundance of bacteria, compared to CRU alone (Fig. 2).” (Lines 396-398)

Point 35: Line 391. Replace "explain the" by "explained why".

Response 35: Thanks for your comment. We have replaced "explain the" with "explained why".

“The positive function of bacteria community of adding FA treatment was significantly increased in the low nitrogen period and explained why CRU+FA increased the yield.” (Lines 409-411)

Point 36: Line 380. Instead of "the 30d to 60d", it would be better to have "in the 30d to 60d period".

Response 36: Thanks for your comment. We have replaced "the 30d to 60d" with "in the 30d to 60d period".

“in the 30d to 60d period, the nutrients of CRU were released in large quantities, and the nutrient concentration during the period belonged to the high nitrogen (Supplementary Fig. S3).” (Lines 400-402)

Point 37: Line 383. "different" rather than "difference".

Response 37: Thanks for your comment. We have replaced "difference" with "different".

“The relative abundance of bacteria on different fertilizer days was significantly different due to the application of FA and varying N concentration in the soil (Fig. 2B).” (Lines 402-404)

Point 38: Line 386 and in several other places. Proteobacteria should consistently be in italics.

Response 38: Thanks for your comment. We have modified in the article.

“We discovered that N fertilizer boosted the relative abundance of Actinobacteria and Proteobacteria (Fig. 2). The positive function of Actinobacteria and Proteobacteria in soils was predicted at the phylum level, and these species have been documented as having a considerable influence on the N cycle [27].” (Lines 405-408)

“In contract, the abundance of Proteobacteria was similar between CRU and CRU+FA treatments on Day 1 but that for CRU+FA became higher than in CRU on the following days.” (Lines 244-246)

“It has been reported that many bacterial communities in Proteobacteria are involved in soil nitrogen cycle, and most of the bacteria community related to nitrification belong to Proteobacteria [33].” (Lines 464-466)

Point 39: Line 391. Insert "was" between "urea" and "applied".

Response 39: Thanks for your comment. We have added "was" between "urea" and "applied".

“When urea was applied to the soil, the nutrients were quickly released, the soil micro-environment reached a high N level.” (Lines 411-413)

Point 40: Line 394. "abundance of advantage of bacteria" doesn't make sense. Needs rewriting.

Response 40: We are very sorry for our unclearly expressing. We have modified the sentence.

“Because the application of FA to high N level of soil significantly decreased the relative abundance of beneficial of bacteria community in the soil,” (Lines 413-415)

Point 41: Sections 4.3 and 4.4. Extensive rewriting is needed in these sections. For example, lines 469–470, "The abundance of Ensifer, Blastococcus and Pseudolabrys all play a role in soil nitrogen fixation capacity [41, 42] were significantly increased" doesn't make sense as it stands.

Response 41: Thank you for your careful modification and your advice for the article. We have modified the sentence.

“After fertilization, the relative abundance of bacterial communities in CRU+FA treatment examples was greater than that in CRU treatment (Fig. 1A, B). The CRU+FA treatment network demonstrated that the co-occurrence of bacterial members in CRU+FA treatment was more intense on various days (Supplementary Fig. S8). According to the network analysis, the bacterial community was more sensitive to FA; species vulnerable to N fertilizer were clustered together, indicating that exogenous perturbations may have a significantly greater effect on the bacterial community by targeting vulnerable species (Supplementary Fig. S9B).” (Lines 418-431)

“The nutrient release of CRU reduced the variety of interaction linkages, which is substantiated by fewer bacterial node connections (Supplementary Fig. S9A). Moreover, the CRU+FA network expanded in number of linkages, indicating that adding the FA may benefit a variety of mutualistic bacteria (Supplementary Fig. S9B). In brief, high N fertilizer may impact community structure stability by increasing mutualistic relationships among bacteria [3]. In N nutrient release of CRU was consistent with the nutrient absorption of wheat in soils, sophisticated ecological linkages between bacteria may protect the integrity of the community structure. The addition of FA, on the other hand, disrupted the original relationship of bacteria, stimulated the growth of dominant bacteria, altered the initial ecological network linkages, and had a synergistic impact.” (Lines 437-448)

“The Ensifer, Blastococcus and Pseudolabrys all play a role in soil nitrogen fixation capacity [41, 42], and the relative abundance of which was significantly increased. The FA contains many active ions, which could promote the growth and nitrogen fixation ability of Ensifer, Blastococcus and Pseudolabrys with organic carbon as a carbon source. Ensifer, Blastococcus and Pseudolabrys could fix inorganic N in soil, and could be biologically fixed to microorganisms in the process of growth and reproduction. Meanwhile, Ensifer, Blastococcus and Pseudolabrys proliferating and expanding could alter the dominant population of soil microorganisms and limit the growth of some alien microorganisms. Wang [27] and Zhou [28] studied the effect of N fertilizer on soil microbial biomass and inorganic N, and obtained that N fertilizer significantly inhibited the abundance of soil microorganisms.” (Lines 496-506)

“This difference may occur that the application of FA could provide a carbon source for the growth of microorganisms, while the nutrients release of CRU were slow and required a certain amount of heat and water. Thus, Ensifer, Blastococcus and Pseudolabrys were unable to directly consume all the inorganic nitrogen of the CRU and fix it to microbial nitrogen. Most of the bacterial communities considerably affected by adding FA treatment had a favorable influence on wheat growth promotion, yield and NUE.” (Lines 511-517)

Point 42: Line 487. "fix", not "fixe".

Response 42: Thanks for your comment. We have replaced "fixe" with "fix".

“Thus, Ensifer, Blastococcus and Pseudolabrys were unable to directly consume all the inorganic nitrogen of the CRU and fix it to microbial nitrogen.” (Lines 514-515)

Point 43: Line 503. The degree symbol is missing.

Response 43: Thanks for your comment. We have added the degree symbol.

“Supplementary Fig. S5: Nitrogen release of the controlled release urea, in water at 25°C (a) and in soil under field condition (b);” (Lines 538-540)

Point 44: Supplementary Table S1. Are the entries the means of 3 replicates? It should be stated what these are. Can standard errors be added?

Response 44: Thanks for your comment. We have modified the Supplementary Table S1.

Supplementary Table S1 The soil nutrients change of the soil microbial incubation, the soil column leaching and the pot experiments.
